# Infant Feeding and Growth Patterns in Japanese Children: A Nationwide Secondary Analysis

**DOI:** 10.3390/nu17223566

**Published:** 2025-11-14

**Authors:** Akinori Moriichi, Erika Kuwahara, Narumi Kato

**Affiliations:** Division of Information for Specific Pediatric Chronic Diseases, Research Institute, National Center for Child Health and Development, Tokyo 157-8535, Japan

**Keywords:** anthropometry, feeding behavior, growth and development, nutritional status, preschool children

## Abstract

**Objectives**: To examine age-specific growth patterns derived from Japanese cross-sectional data according to infants’ feeding and determine whether differences persist through preschool age. **Methods**: We analyzed secondary data from the 2023 National Growth Survey on Preschool Children in Japan, a single-wave nationwide cross-sectional survey. The participants were 8028 singleton, term-born, appropriate-for-gestational-age children aged 0–60 months without major health conditions. The feeding history up to 24 months was reported by parents and categorized as breastfed, formula-fed, or mixed-fed. Anthropometric measurements were obtained at a 1-month postnatal checkup or at checkups arranged for the survey, converted to standard deviation scores using Japanese references, and modeled with growth curves using the Lambda–Mu–Sigma method to summarize cross-sectional distributions by age. The feeding groups were compared at selected ages. **Results**: Breastfed infants were smaller in length/height and weight than formula-fed peers during the first 2 years, with the largest differences in infancy. The mean stature in the feeding groups became similar at older ages; by 60 months, standard deviation scores for stature and weight were comparable across the feeding groups. Head circumference patterns up to 36 months were not different by the feeding category. **Conclusions**: In Japan, early size differences by the feeding group show age-related convergence of cross-sectional group means by preschool, and head circumference patterns are similar across the groups. These findings support breastfeeding as sufficient for long-term growth without unnecessary formula supplementation.

## 1. Introduction

Infant feeding practices are major determinants of child growth and development. Breastfeeding provides multiple short- and long-term benefits, such as protection against infectious diseases, reduced risk of sudden infant death, and positive effects on cognitive development and maternal–infant bonding [1,2,3,4,5,6]. Commercial infant formula is designed to meet infant nutrient requirements when breastfeeding is not possible. Nevertheless, previous studies have reported that formula is associated with more rapid somatic growth in early infancy and a higher risk of overweight or obesity than breastfeeding [7,8,9,10,11]. Previous studies have suggested that infants fed with formula tend to gain more weight and to increase in length more rapidly than breastfed infants during the first year of life, raising concerns about their risk of later overweight and obesity [2,9,10,11,12]. The difference in growth between formula-fed and breastfed infants has been shown to decrease by early childhood in European countries and the United States [8,13,14]. In contrast, Japanese studies have reported that breastfed infants are smaller than formula-fed infants during early infancy, but their subsequent growth has not been well studied, and unlike in other countries, they may not achieve catch-up growth by 3 years of age [15,16].

This study aimed to construct growth curves for Japanese children from birth to 5 years of age according to breastfed, formula-fed, and mixed-fed feeding types using data from an extensive, nationally representative cross-sectional survey. The study objectives were to determine whether anthropometric differences between feeding groups persist into preschool years, to examine sex-specific growth patterns, and to assess whether head circumference trajectories differ by the feeding practice.

This study is important for the following reasons. First, we used a large, nationally representative sample from the 2023 National Growth Survey on Preschool Children, with standardized anthropometric data collected from 0 to 60 months within a single survey wave. This approach enabled age-specific cross-sectional comparisons of body size by feeding group that are rarely available in Japan. Second, using the sex-specific Lambda–Nu–Sigma method with centile modeling, we quantified between-group differences at clinically salient ages (36 and 60 months) in absolute units and standard deviation scores (SDSs), allowing clinicians to interpret the practical magnitude of any differences. Third, the mean values for physical stature across the feeding groups tended to converge with age, which suggested that differences attributable to early nutrition practices were generally small. This finding may help inform family-oriented guidance in the preschool years and individualized public health messaging.

## 2. Methods

### 2.1. Participants

Data were obtained from the 2023 National Infant Growth Survey on preschool children, which was a single-wave, nationwide cross-sectional survey conducted by the Japanese government in 2023. Although national surveys are conducted approximately every 10 years, this analysis used only the 2023 wave. Participants were recruited through two complementary sampling frameworks: (1) maternal healthcare facilities, which routinely collect anthropometric data during 1-month checkups and provide information on neonates; and (2) municipal health examinations conducted in randomly sampled districts for this survey, targeting infants and children aged 2 weeks to <7 years. This dual approach reflects a Japanese cultural practice where neonates generally remain at home until their 1-month checkup at a birth facility. Outings are avoided during this period, which makes participation in health surveys requiring travel difficult. Each child had one observation at the 1-month checkup or the survey examination. The eligibility criteria were as follows: singleton births at ≥37 weeks of gestation, birth weight appropriate for gestational age, and no major congenital anomalies or chronic medical conditions known to affect growth. Children born small-for-gestational age, defined as a birth weight below the 10th percentile for gestational age, or large-for-gestational age, defined as a birth weight above the 90th percentile for gestational age, were excluded to minimize heterogeneity related to perinatal growth restriction or overgrowth [17]. Small-for-gestational-age infants have unique postnatal growth patterns and frequently receive modified feeding for medical reasons. Therefore, their inclusion could confound associations between the feeding category and body size. Consequently, we restricted our analyses to term-appropriate-for-age singletons.

### 2.2. Feeding Group Classification

The participants were categorized into three feeding groups: breastfed, formula-fed, and mixed-fed groups. Caregivers reported monthly feeding histories from birth to 24 months, and indicated whether the infant received breast milk and/or formula each month. Infants reported to have never received formula milk up to 24 months of age were classified as breastfed. The formula-fed group was defined as infants who had never been breastfed during the first 24 months and those whose duration of formula feeding was at least twice as long as the duration of breastfeeding. The month-by-month feeding status from birth to 24 months for each feeding group is shown in Table A1, Table A2 and Table A3. Participants who did not meet the criteria for the breastfed or formula-fed groups were assigned to the mixed-fed group. The definition of the breastfed group in this study does not correspond precisely to the WHO definition of exclusive breastfeeding, which excludes all other liquids and solids, except oral rehydration solution or drops and syrups of vitamins, minerals, or medicines [18].

### 2.3. Anthropometric Measures

Anthropometric data at birth were retrieved from medical records or maternal and child health handbooks, and those at birth and at the time of the survey were measured directly, covering ages from birth to 60 months. Length was measured in the supine position before independent walking, and standing height was measured thereafter. Weight was recorded using calibrated scales, and head circumference was measured up to 36 months of age using non-stretchable measuring tapes at the maximum occipito-frontal circumference. Age and sex were recorded, and all measures were converted to SDSs relative to the Japanese reference growth standards [19]. The total number of measurements available for each parameter is shown in Table A4.

### 2.4. Growth Curve Modeling

Sex- and feeding-specific growth curves for height/length, weight, and head circumference were generated. Growth curves were constructed using the Lambda–Mu–Sigma method [20,21], which summarizes age-specific distributions of anthropometric measures in terms of the median, coefficient of variation, and skewness parameter. These three parameters were modeled as smooth functions of age using penalized splines within the Generalized Additive Models for Location, Scale, and Shape framework in R [22], which enabled the estimation of smoothed centile curves suitable for clinical and epidemiological interpretation.

### 2.5. Statistical Analysis

Baseline characteristics are summarized using descriptive statistics. Continuous variables are expressed as the mean with standard deviation (SD), and were compared between three groups by one-way analysis of variance with Bonferroni-adjusted post hoc comparisons. The 50th percentile (0 SD) somatometric values obtained from the growth curves at 36 and 60 months were also compared directly to assess potential differences between feeding groups. Statistical significance was defined as a two-tailed *p* value < 0.05. All analyses were performed using SPSS version 30 (IBM Corp., Armonk, NY, USA) and R version 4.5.0 (R Foundation for Statistical Computing, Vienna, Austria).

## 3. Results

### 3.1. Study Population

A total of 8028 children were included in the final analysis, with 1743 (21.7%) in the breastfed group, 1602 (20.0%) in the formula-fed group, and 4683 (58.3%) in the mixed-fed group (Figure 1). The month-by-month distribution of the feeding mode in each group is shown in Table A1, Table A2 and Table A3, respectively. Table 1 shows the background characteristics of the participants. Although some background factors were significantly different between the groups, no clinically meaningful differences were identified. There was no difference in the timing of starting weaning between the groups.

### 3.2. Height/Length and Weight Patterns

Figure A1 shows the growth curves of height/length and weight by nutritional type for boys and girls. Figure 2 illustrates the age-specific SDSs for 0 SD of height/length in each group, relative to the Japanese standard growth curve. The SDSs of height/length in breastfed children of both sexes remained lower than those in the other groups. However, at 60 months, these values were similar in all groups.

Figure 3 shows the age-specific SDSs for 0 SD weight values in each group, relative to the Japanese standard curve. Although the values of breastfed infants of both sexes were generally lower than those of formula-fed infants throughout much of the study period, the between-group differences were small by 60 months.

### 3.3. Head Circumference Patterns

Figure A2 shows the growth curves of head circumference by sex and nutrition. The growth curves are available up to 36 months because head circumference was measured up to approximately 3 years of age. Figure 4 shows the age-specific SDSs for 0 SD head circumference values in each group, relative to the Japanese standard curve. There was little difference in the SDSs of head circumference between the groups for both sexes.

### 3.4. Anthropometric Values at 36 and 60 Months

Table 2 shows the estimated 0 SD values for height/length, weight, and head circumference at 36 and 60 months, which were calculated from the sex-specific growth curves constructed in this study. The corresponding SDSs were derived relative to the Japanese reference growth standards. These values are shown separately for boys and girls by feeding group. The absolute differences in anthropometric measurements between breastfed children and those who received other feeding methods were minimal. Except for weight in 60-month-old boys, these differences were within the expected measurement error range. The corresponding differences in SDSs were also within ±0.3 SD, which is considered clinically negligible, indicating no meaningful differences between the groups.

## 4. Discussion

### 4.1. Main Findings

In this study, growth curves were created for each nutritional method in a cross-sectional national survey and used to summarize age-specific differences in body size for the Japanese population. Breastfed infants tended to be smaller in length and lower in weight at younger ages than formula-fed infants. There was no difference in head circumference between the nutritional methods. No clinically significant difference in body size according to the feeding method was observed at 36 or 60 months.

Growth differences related to feeding practices have been reported to resolve by 1–2 years of age [8,13,14]. A longer duration of breastfeeding is associated with lower z-scores at 12 months, but this difference disappears by 24 months [8]. From approximately 2–3 months of age, formula-fed infants gain more lean body mass and become larger than breastfed infants. Nevertheless, variations in body composition are no longer evident after 2 years of age [1,13,23].

Although reports from Japan have shown that breastfed infants are smaller in size than formula-fed infants [16,24], there have been few reports on whether this difference in body size disappears after infancy. In a recent Japanese cohort study conducted in a single prefecture, breastfed children were shorter in stature than formula-fed children at 3 years of age, but differences attributable to infant feeding practices were no longer evident by 6 years of age [15]. Previous Japanese studies reported that breastfed infants followed up to 24 months showed length and weight below the standard values [24], and that breastfed infants were shorter than formula-fed infants at 36 months [15,16]. However, by 6 years of age and thereafter, no significant differences in stature remained between feeding groups [15]. These findings suggest age-related convergence of cross-sectional group means by preschool age in Japan.

We constructed feeding type-specific growth curves by extracting healthy singleton, appropriate-for-gestational-age, term-born children from a nationally representative survey of the pediatric population obtained through cluster random sampling. At 2 years of age, breastfed children tended to be shorter than those in the other feeding groups in both sexes. However, at 5 years of age, between-group differences were small, and the maximum absolute difference in 0 SD height between the breastfed group and the other groups was ≤0.7 cm. This intergroup difference was within the range of measurement error (0.6–2.0 cm) [25,26,27], and the SDS difference was <0.3 SD, which indicated a clinically negligible difference.

Importantly, head circumference, a proxy for brain development, was not affected by the feeding type in this study. This finding is consistent with reports suggesting that cognitive outcomes are not compromised in breastfed infants despite slower somatic growth [2,28]. These results support flexible feeding choices based on family circumstances while acknowledging the modest effect of formula on physical size.

Previous studies have suggested a potential association between formula feeding and an increased risk of later obesity [2,9,10,11,12]. In Japan, cohort data have indicated that formula-fed children may show a higher prevalence of obesity at 15 years of age than those who are breastfed [29]. Infant formula generally contains more protein than human milk, and this higher protein intake has been suggested to increase the risk of rapid weight gain during early infancy [11,30]. Rapid weight gain in infancy is associated with an increased risk of overweight and obesity [10,31]. Moreover, formula feeding might be associated with increased body mass index in school-age children, independent of whether rapid weight gain occurs [10]. In our study, we did not assess adiposity or metabolic outcomes, and no causal inference can be made. Breastfed infants are generally smaller in stature during infancy and early childhood, but excessive supplementation with formula is not recommended. Accordingly, careful assessment of growth patterns and provision of appropriate nutritional guidance are warranted for breastfed infants.

### 4.2. Strengths and Limitations

Strengths of the study include the following. The data used in this study were obtained from the National Growth Survey on Preschool Children. This survey applies random sampling of districts across Japan to minimize potential biases in the pediatric population, thereby providing highly representative data. Additionally, the growth curves were constructed exclusively from healthy participants, defined as singleton, term-born children, with exclusion of those presenting any underlying conditions that could affect growth.

This study has several limitations that should be acknowledged. First, because the growth curves were derived from cross-sectional data, evaluating individual longitudinal variations in growth trajectories was not possible. Second, infants born small-for-gestational age were excluded, although such children may follow distinct growth patterns characterized by catch-up growth. As a result of excluding small-for-gestational-age and large-for-gestational-age infants, the generalizability of our findings is limited to healthy, term appropriate-for-gestational-age children. Therefore, dedicated analyses are needed for populations with perinatal growth restriction or overgrowth. Third, the number of exclusively formula-fed infants was relatively small, and therefore, children whose formula feeding duration was more than twice that of breastfeeding were classified into the formula-fed group. This approach may have underestimated the effects of formula feeding. However, during the first 12 months of life, when the influence of milk is considered to be greatest, children who were more frequently formula-fed tended to be taller, suggesting that any potential underestimation was unlikely to be substantial. Fourth, infants categorized as breastfed may not have fully satisfied the WHO criteria for exclusive breastfeeding. Finally, slight differences were observed in maternal age and birth size between the groups, and detailed information on maternal lifestyle or socioeconomic status was not available. As a result, unmeasured baseline characteristics may have influenced feeding practices and growth outcomes. Endocrine markers, dietary macronutrient information (including protein), and body composition measurements were not collected. Therefore, mechanistic explanations and inferences regarding the risk of later obesity are beyond the scope of this study.

Despite these limitations, the large sample size and use of nationally representative data strengthen the validity of our findings and provide meaningful insights into growth patterns according to the feeding type in Japan. In this nationwide cross-sectional survey, we found that the differences in height and weight observed during infancy, depending on the method of nutritional intake, were similar between the feeding groups by the time of preschool age. In contrast, the pattern of head circumference was equivalent between the groups at all ages. These findings provide evidence that breastfeeding is compatible with appropriate growth. Our findings suggest that clinicians should support the recommendation that additional formula feeding is not advised for healthy infants. Additionally, we recommend that policymakers support breastfeeding and regular physical measurements for monitoring, which facilitate the reduction in unnecessary formula milk feeding, contributing to improved communication within the family. Furthermore, our findings indicate that caregivers need to be aware that the smaller physique observed in infants fed breast milk than in those fed a formula is common and generally not an issue to be concerned about because it usually has limited clinical significance.

Future research is necessary to investigate mechanistic hypotheses, such as the relationships between body composition, major nutrient intake, and growth patterns, in longitudinal cohorts from infancy to school age, as well as the association between future obesity and endocrine markers.

## 5. Conclusions

Breastfed Japanese infants tended to show a slightly smaller stature than formula-fed infants during early childhood, but their height becomes nearly equivalent by approximately 5 years of age. Moreover, head circumference growth—an indicator closely associated with neurodevelopment—is similar among the feeding methods. Recommendations for supplemental nutrition should be cautiously made because of the likelihood of subsequent catch-up growth.

## Figures and Tables

**Figure 1 nutrients-17-03566-f001:**
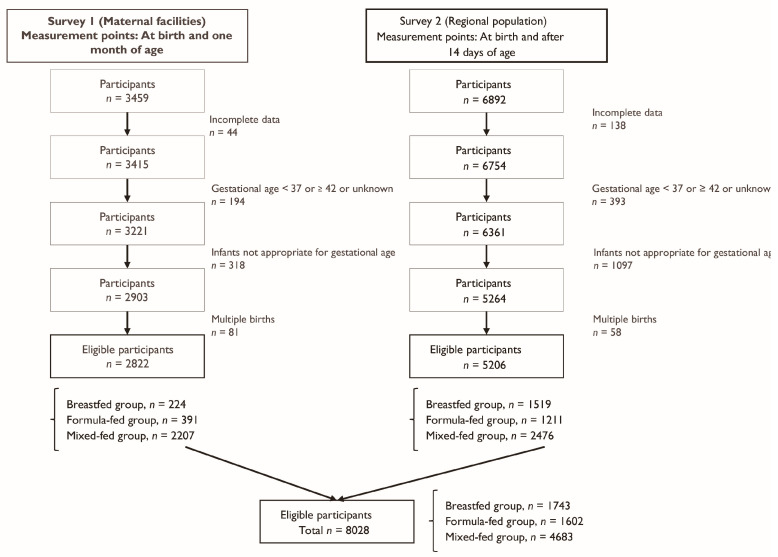
Flow diagram of participants’ enrollment.

**Figure 2 nutrients-17-03566-f002:**
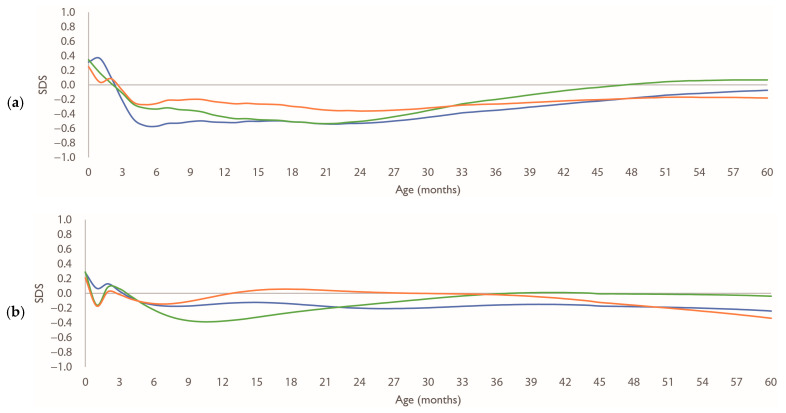
Growth patterns of height/length SDS by sex. (**a**,**b**) age-specific SDS patterns of 0 SD values calculated using the LMS method, for boys and girls, respectively, relative to Japanese standard growth curves. Blue line: breastfed group; green line: mixed-fed group; orange line: formula-fed group. Abbreviations: LMS, Lambda–Mu–Sigma; SD, standard deviation; SDS, standard deviation scores.

**Figure 3 nutrients-17-03566-f003:**
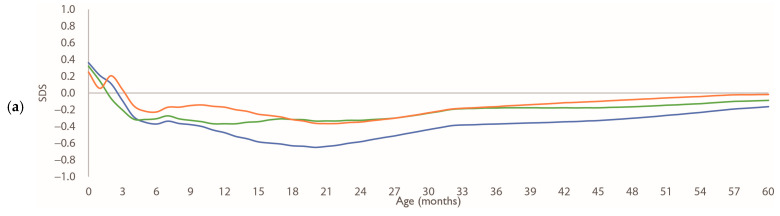
Growth patterns of weight SDS by sex. (**a**,**b**) age-specific SDS patterns of 0 SD values calculated using the LMS method, for boys and girls, respectively, relative to Japanese standard growth curves. Blue line: breastfed group; green line: mixed-fed group; orange line: formula-fed group. Abbreviations: LMS, Lambda–Mu–Sigma; SD, standard deviation; SDS, standard deviation scores.

**Figure 4 nutrients-17-03566-f004:**
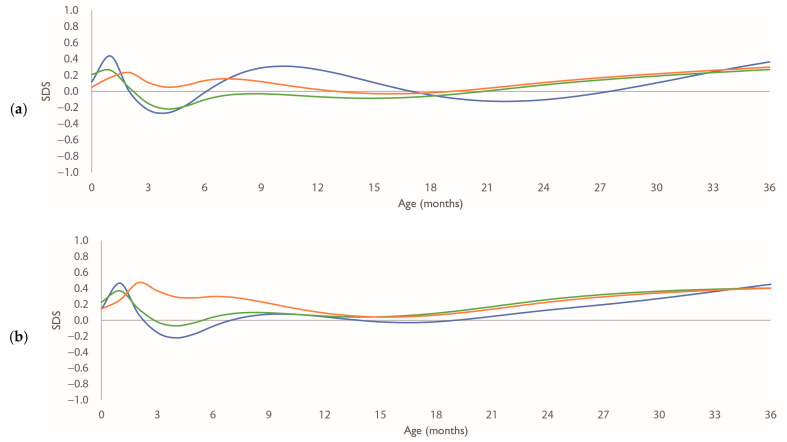
Growth patterns of head circumference SDS by sex. (**a**,**b**) age-specific SDS patterns of 0 SD values calculated using the LMS method, for boys and girls, respectively, relative to Japanese standard growth curves. Blue line: breastfed group; green line: mixed-fed group; orange line: formula-fed group. Abbreviations: LMS, Lambda–Mu–Sigma; SD, standard deviation; SDS, standard deviation scores.

**Table 1 nutrients-17-03566-t001:** Baseline characteristics of the participants.

Variables	Feeding Type	Boys				Girls			
		*n*	Mean	SD	*p*	*n*	Mean	SD	*p*
Maternal age, years	Breastfed	895	31.8 ^†‡^	(4.7)	<0.001	848	31.5 ^‡^	(4.6)	<0.001
	Formula-fed	811	30.7 ^†§^	(5.6)		791	31.5 ^§^	(5.8)	
	Mixed-fed	2354	32.5 ^‡§^	(5.0)		2329	32.5 ^‡§^	(5.1)	
Gestational age, weeks	Breastfed	895	39.3	(1.0)	0.531	848	39.4 ^†^	(1.1)	0.032
	Formula-fed	811	39.3	(1.1)		791	39.3 ^†^	(1.1)	
	Mixed-fed	2354	39.3	(1.1)		2329	39.4	(1.1)	
Birth weight, g	Breastfed	895	3131	(285)	0.019	848	3037	(276)	0.168
	Formula-fed	811	3108 ^§^	(291)		791	3012	(283)	
	Mixed-fed	2354	3142 ^§^	(301)		2329	3032	(298)	
Birth length, cm	Breastfed	894	49.6	(1.7)	0.035	848	49.1 ^†^	(1.7)	0.017
	Formula-fed	810	49.5 ^§^	(1.7)		788	48.9 ^†^	(1.8)	
	Mixed-fed	2340	49.7 ^§^	(1.7)		2302	49.0	(1.8)	
Head circumference at birth, cm	Breastfed	893	33.7	(1.2)	0.007	843	33.3	(1.7)	0.307
	Formula-fed	808	33.6 ^§^	(1.3)		784	33.2	(1.2)	
	Mixed-fed	2328	33.8 ^§^	(1.3)		2290	33.3	(1.2)	
Age at introduction of complementary foods, months	Breastfed	701	5.6	(1.2)	0.038	674	5.6	(1.1)	0.207
	Formula-fed	531	5.5	(1.1)		586	5.6	(1.1)	
	Mixed-fed	1074	5.5	(0.9)		1076	5.5	(0.9)	

*p*-values shown in the table were determined by one-way analysis of variance. Symbols indicate pairwise post hoc differences with Bonferroni correction (adjusted *p* < 0.05): ^†^ breastfed vs. formula-fed; ^‡^ breastfed vs. mixed-fed; and ^§^ formula-fed vs. mixed-fed. Abbreviations: SD, standard deviation.

**Table 2 nutrients-17-03566-t002:** Values of somatometric measurements at 36 and 60 months by sex and feeding type.

Sex	Age	Measurement	Feeding Type						Maximum Absolute Difference Between the Breastfed and Other Feeding Type	
			Breastfed		Formula-Fed		Mixed-Fed			
			0 SD	SDS	0 SD	SDS	0 SD	SDS	ΔSD	ΔSDS
Boys	36 months	Height/length (cm)	92.1	−0.35	92.4	−0.27	92.6	−0.20	0.5	0.15
		Weight (kg)	13.3	−0.16	13.5	−0.02	13.5	−0.01	0.2	0.15
		Head circumference (cm)	50.0	0.36	49.9	0.30	49.9	0.30	0.1	0.06
	60 months	Height/length (cm)	106.4	−0.07	105.9	−0.18	107.0	0.07	0.6	0.14
		Weight (kg)	16.9	−0.24	16.7	−0.34	17.4	−0.04	0.5	0.20
Girls	36 months	Height/length (cm)	91.0	−0.37	91.7	−0.16	91.6	−0.18	0.7	0.21
		Weight (kg)	12.9	−0.11	12.9	−0.11	13.1	0.02	0.2	0.13
		Head circumference (cm)	49.1	0.45	49.0	0.40	49.0	0.40	0.1	0.05
	60 months	Height/length (cm)	105.5	−0.16	106.1	−0.02	105.8	−0.09	0.6	0.14
		Weight (kg)	16.5	−0.30	16.5	−0.32	16.7	−0.18	0.2	0.12

Values for each group show 0 SDs at 36 and 60 months of age, which were calculated from curves constructed using the Lambda–Mu–Sigma method, with SDSs relative to Japanese standard growth curves. Abbreviations: SD, standard deviation; SDS, standard deviation score.

## Data Availability

The datasets presented in this article are not readily available because they were provided by the Children and Families Agency under usage restrictions. Requests for access to the datasets should be directed to the Children and Families Agency, Japan.

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
