# Peer review of "Infant Feeding and Growth Patterns in Japanese Children: A Nationwide Secondary Analysis"

_nutrients, 2025, doi:10.3390/nu17223566_

Round 1

Reviewer 1 Report

Comments and Suggestions for Authors

The main question of this manuscript is the persistent-"eternal" question, how do you feed a baby in the critical period (60 months) in order to suceed a healthier adult life and longevity.
There is an originality in this manuscript but the case (Charts, etc.) gives also answers in a country of one hundred million inhabitans.
 I considered that some results like the similar head circumference growth between groups and what that it means, confirms the existing possibilities for feeding and developing an infant.
 Methodology: to give more informations for babies in the formula-fed group to those that the duration of formula feeding was at least twice as long as the duration of breast feeding (time, numbers)
Conclusions are consistent, references are appropriate and i have no comments on the tables and figures.

Author Response

Reviewer’s comment 1:

The main question of this manuscript is the persistent-“eternal” question, how do you feed a baby in the critical period (60 months) in order to suceed a healthier adult life and longevity.
There is an originality in this manuscript but the case (Charts, etc.) gives also answers in a country of one hundred million inhabitans.
I considered that some results like the similar head circumference growth between groups and what that it means, confirms the existing possibilities for feeding and developing an infant.

Authors' response 1:

We thank the reviewer for the positive evaluation of the study’s aims, originality, and national relevance. We agree that the similarity in head circumference patterns across the feeding groups is informative in this context.

Reviewer’s comment 2:

Methodology: to give more informations for babies in the formula-fed group to those that the duration of formula feeding was at least twice as long as the duration of breast feeding (time, numbers). 
Conclusions are consistent, references are appropriate and i have no comments on the tables and figures.

Authors' response 2:

We appreciate the reviewer’s useful comment. We have mentioned in the Methods section that details on breastfeeding and the formula feeding duration are provided in Tables A1-1, A1-2, and A1-3, and have added the proportions for each group to the Results section.

Changes to the manuscript
Methods section (Feeding group classification):

“The participants were categorized into three feeding groups: breastfed, formula-fed, and mixed-fed groups. Caregivers reported monthly feeding histories from birth to 24 months, and indicated whether the infant received breast milk and/or formula each month. Infants reported to have never received formula milk up to 24 months of age were classified as breastfed. The formula-fed group was defined as infants who had never been breastfed during the first 24 months and those whose duration of formula feeding was at least twice as long as the duration of breastfeeding. The month-by-month feeding status from birth to 24 months for each feeding group is shown in Tables A1-1 to A1-3.”

Results section (Study population):

“A total of 8,028 children were included in the final analysis, with 1,743 (21.7%) in the breastfed group, 1,602 (20.0%) in the formula-fed group, and 4,683 (58.3%) in the mixed-fed group (Figure 1). The month-by-month distribution of the feeding mode in each group is shown in Tables A1-1, A1-2, and A1-3, respectively.”

Reviewer 2 Report

Comments and Suggestions for Authors

The authors have produced an interesting study on the effects of breastfeeding among Japanese infants and young children. The data are rich and the methods seem sound, though I am not an expert in the specific statistical tests that are applied. The study could be improved with attention to the following considerations. 

  1. This study has a great deal of significance (unique contributions), which seems to include data that are richer than much of what has been used in previous publications. Also, the argument for foundational development in infancy and early childhood with influence on later life is noteworthy here. I recommend adding a paragraph to the introduction specifying the significance of this research. The authors should make the strongest possible case for their study. For example, this study can address a gap in developmental trajectories regarding body size. I'd suggest explicitly stating, "This study is significant for the following reasons. First, ... Second, ..."
  2. The methods need greater clarity. If this is a cross-sectional study, how do the researchers know about diminishing developmental differences over time? That conclusion seems like it would require panel data with the same individuals tracked over time. Are repeated cross-sectional data on different groups being used rather than tracking the same individuals over time? The abstract reads like panel data are needed to draw the conclusions shared despite the abstract mentioning cross-sectional data. For developmental attributes over time, do the data feature linked records that permit specific participants to be tracked? And how can that be done at 5 years of age if the government collects these data every 10 years? This seems like a combination of cross-sectional data and measure-specific follow-up developmental data collected from the same infants/children, the latter focused on growth outcomes. If my terminology is wrong here, feel free to not follow my lead. But please provide more detail on the data that are not exactly cross-sectional in nature and revise the prose accordingly. Leaving it at cross-sectional in the abstract and some areas of the manuscript invites confusion.   
  3. The exclusion of infants born small for gestational age is understandable to create comparable subgroups, but could be better justified with an additional sentence or two. 
  4. How about adding two paragraphs at the end? One of these could provide practical recommendations for policymakers, medical professionals, and even parents. The other could outline directions for future research to address limitations of this study or unanswered questions. 
  5. Small consideration, but I generally see spaces on either side of a <, like this: p < 0.001. The same applies to = and other mathematical symbols. 

This is a generally well done study with some room for refinement. 

Author Response

Reviewer’s comment 1:

The authors have produced an interesting study on the effects of breastfeeding among Japanese infants and young children. The data are rich and the methods seem sound, though I am not an expert in the specific statistical tests that are applied. The study could be improved with attention to the following considerations.

Authors' response 1:

We sincerely thank the reviewer for the positive assessment of our research question, data, and overall approach.

Reviewer’s comment 2:

This study has a great deal of significance (unique contributions), which seems to include data that are richer than much of what has been used in previous publications. Also, the argument for foundational development in infancy and early childhood with influence on later life is noteworthy here. I recommend adding a paragraph to the introduction specifying the significance of this research. The authors should make the strongest possible case for their study. For example, this study can address a gap in developmental trajectories regarding body size. I'd suggest explicitly stating, "This study is significant for the following reasons. First, ... Second, ..."

Authors' response 2:

We appreciate the reviewer’s suggestion. We have added a dedicated paragraph to the end of the Introduction section that clearly states the study’s importance.

Changes to the manuscript
Introduction (final paragraph):

“This study is important for the following reasons. First, we used a large, nationally representative sample from the 2023 National Growth Survey on Preschool Children, with standardized anthropometric data collected from 0 to 60 months within a single survey wave. This approach enabled age-specific cross-sectional comparisons of body size by feeding group that are rarely available in Japan. Second, using the sex-specific Lambda-Nu-Sigma method with centile modeling, we quantified between-group differences at clinically salient ages (36 and 60 months) in absolute units and standard deviation scores (SDSs), allowing clinicians to interpret the practical magnitude of any differences. Third, the mean values for physical stature across the feeding groups tended to converge with age, which suggested that differences attributable to early nutrition practices were generally small. This finding may help inform family-oriented guidance in the preschool years and individualized public health messaging.”

Reviewer’s comment 3:

The methods need greater clarity. If this is a cross-sectional study, how do the researchers know about diminishing developmental differences over time? That conclusion seems like it would require panel data with the same individuals tracked over time. Are repeated cross-sectional data on different groups being used rather than tracking the same individuals over time? The abstract reads like panel data are needed to draw the conclusions shared despite the abstract mentioning cross-sectional data. For developmental attributes over time, do the data feature linked records that permit specific participants to be tracked? And how can that be done at 5 years of age if the government collects these data every 10 years? This seems like a combination of cross-sectional data and measure-specific follow-up developmental data collected from the same infants/children, the latter focused on growth outcomes. If my terminology is wrong here, feel free to not follow my lead. But please provide more detail on the data that are not exactly cross-sectional in nature and revise the prose accordingly. Leaving it at a cross-sectional level in the abstract and in some areas of the manuscript invites confusion.

Authors' response 3:

We thank the reviewer for pointing out that better clarity of the methods is required. We agree that our original wording could be interpreted as within-child changes. The 2023 national survey involved a single-wave, cross-sectional dataset. Each child had one anthropometric observation between 0 and 60 months plus measurement variables at birth. There were no linked repeated measures for the same child beyond birth versus the survey time. Therefore, we did not use panel/longitudinal data, and we did not pool multiple survey years as repeated cross-sections.

The reference to patterns “over time” referred to age-specific mean differences in a single cross-section, not to within-person trajectories. We estimated age-specific distributions by fitting sex-specific Lambda-Mu-Sigma centile curves across ages in the same 2023 cross-section and then compared feeding groups at the same age. When group differences were smaller at older ages than at younger ages in this cross-section, we have now described this as age-related convergence of cross-sectional group means,” not “diminishing differences over time.”

In the 2023 survey, children aged 0–6 years were randomly sampled across multiple age cohorts at a single time point. Therefore, our results were based on a single-year, cross-sectional dataset. Although the government survey is conducted every 10 years with different individuals sampled at each wave, the present study analyzed data only from the 2023 survey.

We have removed and rewritten language suggesting within-child changes (e.g., “trajectory” and “catch-up growth”) and replaced it with cross-sectional phrasing. We have also added a sentence stating that within-child changes cannot be inferred. We have added explicit statements of the cross-sectional design to the Abstract and Methods section, clarified interpretation in the Results section, and refined the Discussion section.

Changes to the manuscript
Title

“Infant Feeding and Growth Patterns in Japanese Children: A Nationwide Secondary Analysis”

Abstract (Objectives)

“To examine age-specific growth patterns derived from Japanese cross-sectional data according to infants’ feeding and determine whether differences persist through preschool age.”

Abstract (Methods)

“We analyzed secondary data from the 2023 National Growth Survey on Preschool Children in Japan, a single-wave nationwide cross-sectional survey.”

“Anthropometric measurements were obtained at a 1-month postnatal checkup or at checkups arranged for the survey, converted to standard deviation scores using Japanese references, and modeled with growth curves using the Lambda-Mu-Sigma method to summarize cross-sectional distributions by age. The feeding groups were compared at selected ages.”

Abstract (Results)

“The mean stature in the feeding groups became similar at older ages; by 60 months, standard deviation scores for stature and weight were comparable across the feeding groups.”

Abstract (Conclusions)

“In Japan, early size differences by the feeding group show age-related convergence of cross-sectional group means by preschool, and head circumference patterns are similar across the groups.”

Methods (Participants)

“Data were obtained from the 2023 National Infant Growth Survey on preschool children, which was a single-wave, nationwide cross-sectional survey conducted by the Japanese government in 2023. Although national surveys are conducted approximately every 10 years, this analysis used only the 2023 wave.”

“Each child had one observation at the 1-month checkup or the survey examination.”

Results (Height/length and weight patterns)

“Figure 2 illustrates the age-specific SDSs for 0 SD of height/length in each group, relative to the Japanese standard growth curve.”

“Figure 3 shows the age-specific SDSs for 0 SD weight values in each group, relative to the Japanese standard curve. Although the values of breastfed infants of both sexes were generally lower than those of formula-fed infants throughout much of the study period, the between-group differences were small by 60 months.”

Results (Head circumference patterns)

“Figure 4 shows the age-specific SDSs for 0 SD head circumference values in each group, relative to the Japanese standard curve.”

Results (Anthropometric values at 36 and 60 months)

“Table 2 shows the estimated 0 SD values for height/length, weight, and head circumference at 36 and 60 months, which were calculated from the sex-specific growth curves constructed in this study.”

Discussion (Main findings)

“In this study, growth curves were created for each nutritional method in a cross-sectional national survey, and used to summarize age-specific differences in body size for the Japanese population.”

“Breastfed infants tended to be smaller in length and lower in weight at younger ages than formula-fed infants.”

“These findings suggest age-related convergence of cross-sectional group means by preschool age in Japan.”

“We constructed feeding type-specific growth curves by extracting healthy singleton, appropriate-for-gestational-age, term-born children from a nationally representative survey of the pediatric population obtained through cluster random sampling.”

“However, at 5 years of age, between-group differences were small, and the maximum absolute difference in 0 SD height between the breastfed and other groups was ≤ 0.7 cm.”

“Accordingly, careful assessment of growth patterns and provision of appropriate nutritional guidance are warranted for breastfed infants.”

Reviewer’s comment 4:

The exclusion of infants born small for gestational age is understandable to create comparable subgroups, but could be better justified with an additional sentence or two.

Authors' response 4:

We agree with the reviewer and have clarified the rationale for exclusion. Infants born small-for-gestational age show distinct postnatal growth dynamics (e.g., early catch-up and altered adiposity trajectories) that are not attributable solely to feeding practices and frequently undergo clinician-directed feeding modifications. Therefore, including small-for-gestational age infants could introduce confounding by indication and inflate between-group variance. To isolate feeding-related differences in otherwise healthy term children, we restricted analyses to appropriate-for-gestational age singletons and excluded children born small-for-gestational age and large-for-gestational age. We have also clarified the resulting generalizability in the imitations part of the Discussion section.

Changes to the manuscript

Methods (Participants)

“Small-for-gestational-age infants have unique postnatal growth patterns and frequently receive modified feeding for medical reasons. Therefore, their inclusion could confound associations between the feeding category and body size. Consequently, we restricted our analyses to term appropriate-for-age singletons.”

Discussion (Strengths and limitations)

“As a result of excluding small-for-gestational-age and large-for-gestational-age infants, generalizability of our findings is limited to healthy, term appropriate-for-gestational-age children. Therefore, dedicated analyses are needed for populations with perinatal growth restriction or overgrowth.”

Reviewer’s comment 5:

How about adding two paragraphs at the end? One of these could provide practical recommendations for policymakers, medical professionals, and even parents. The other could outline directions for future research to address limitations of this study or unanswered questions.

Authors' response 5:

We thank the reviewer for the helpful suggestion. We have added two paragraphs at the end of the Discussion section to provide practical recommendations for clinicians, policymakers, and caregivers, and directions for future research.

Changes to the manuscript
Discussion

“ In this nationwide cross-sectional survey, we found that the differences in height and weight observed during infancy, depending on the method of nutritional intake, were similar between the feeding groups by the time of preschool age. In contrast, the pattern of head circumference was equivalent between the groups at all ages. These findings provide evidence that breastfeeding is compatible with appropriate growth. Our findings suggest that clinicians should support the recommendation that additional formula feeding is not advised for healthy infants. Additionally, we recommend that policymakers support breastfeeding and regular physical measurements for monitoring, which facilitate the reduction of unnecessary formula milk feeding, contributing to improved communication within the family. Furthermore, our findings indicate that caregivers need to be aware that the smaller physique observed in infants fed breast milk than in those fed a formula is common and generally not an issue to be concerned about because it usually has limited clinical significance.

Future research is necessary to investigate mechanistic hypotheses, such as the relationships between body composition, major nutrient intake, and growth patterns, in longitudinal cohorts from infancy to school age, as well as the association between future obesity and endocrine markers.”

Reviewer’s comment 6:

Small consideration, but I generally see spaces on either side of a <, like this: p < 0.001. The same applies to = and other mathematical symbols.
This is a generally well done study with some room for refinement.

Authors' response 6:

We thank the reviewer for this helpful stylistic suggestion. In accordance with standard statistical reporting conventions, we have revised the formatting of all P values and other mathematical expressions throughout the manuscript to include spaces on either side of symbols (e.g., “P < 0.001”). These changes affect formatting only and do not alter any of the reported results.

Reviewer 3 Report

Comments and Suggestions for Authors

Infant Feeding and Growth Trajectories in Japanese Children:  A Nationwide Secondary Analysis

Akinori Moriichi, Erika Kuwahara and Narumi Kato

General comments

The investigators analyzed secondary data from the 2023 National Growth Survey on Preschool Children in Japan. Anthropometric measurements were obtained at a 1-month postnatal checkup or at checkups arranged for the survey. Breastfed infants were smaller in length/height and weight than formula-fed peers during the first 2 years, with the largest differences in infancy. The investigators showed that breastfed Japanese children are smaller in early life but achieve catch-up growth by 5 years.

In the paper´s introduction the authors discuss their concern of increased risk of obesity by formula feeding associated with rapid postnatal growth as opposed to breastfeeding. In the discussion, they cite a recent Japanese study showing that formula-fed children have a higher prevalence of obesity at 15 years of age than those who were breastfed.

Kadowaki T, Matsumoto N, Suzuki E, Mitsuhashi T, Takao S, Yorifuji T. Breastfeeding at 6 months of age had a positive impact on overweight and obesity in Japanese adolescents at 15 years of age. Acta Paediatr. 2023 Jan;112(1):106-114. doi: 10.1111/apa.16551.

However, the discussion does not provide any idea how their findings might relate formula feeding to an increased risk of obesity.

Obviously, crude anthropometric data like weight measurement does not provide answers for body mass composition, distribution of lean and fat mass, relative distribution of white and beige/brown adipose tissue and white/brown adipose tissue stem cell fate decisions.

The authors could improve their DISCUSSION considering that formula feeding is associated with higher blood levels of IGF-1, which accelerates postnatal growth (rapid weight gain).

Socha P, Hellmuth C, Gruszfeld D, Demmelmair H, Rzehak P, Grote V, Weber M, Escribano J, Closa-Monasterolo R, Dain E, Langhendries JP, Riva E, Verduci E, Koletzko B; European Childhood Obesity Trial Study Group. Endocrine and Metabolic Biomarkers Predicting Early Childhood Obesity Risk. Nestle Nutr Inst Workshop Ser. 2016;85:81-8. doi: 10.1159/000439489.

Importantly, IGF-1 also promotes adipogenesis by a lineage bias of endogenous adipose stem/progenitor cells. In fact, IGF-1 attenuates Wnt/β-catenin signaling by activating Axin2/PPARγ pathways in cells of the stromal vascular fraction.

Hu L, Yang G, Hägg D, Sun G, Ahn JM, Jiang N, Ricupero CL, Wu J, Rodhe CH, Ascherman JA, Chen L, Mao JJ. IGF1 Promotes Adipogenesis by a Lineage Bias of Endogenous Adipose Stem/Progenitor Cells. Stem Cells. 2015 Aug;33(8):2483-95. doi: 10.1002/stem.2052.

Thus, IGF-1-mediated acceleration of growth by formula feeding may be an indirect indicator of IGF-1-driven extension of adipocyte stem cells and lifelong obesity risk.

A further limitation of this study that should be mentioned is the missing information of total daily protein intake by the three feeding groups, as the amount of protein intake determines circulatory IGF-1 levels and FTO expression controlling both growth trajectories and adipogenesis.

Melnik BC, Weiskirchen R, John SM, Stremmel W, Leitzmann C, Weiskirchen S, Schmitz G. White Adipocyte Stem Cell Expansion Through Infant Formula Feeding: New Insights into Epigenetic Programming Explaining the Early Protein Hypothesis of Obesity. Int J Mol Sci. 2025 May 8;26(10):4493. doi: 10.3390/ijms26104493.

The reviewer wonders why the authors did not discuss recent publications on rapid weight gain by formula feeding and obesity risk that might substantiate their discussion.

Appleton J, Russell CG, Laws R, Fowler C, Campbell K, Denney-Wilson E. Infant formula feeding practices associated with rapid weight gain: A systematic review. Matern Child Nutr. 2018 Jul;14(3):e12602. doi: 10.1111/mcn.12602.

Dharod JM, McElhenny KS, DeJesus JM. Formula Feeding Is Associated with Rapid Weight Gain between 6 and 12 Months of Age: Highlighting the Importance of Developing Specific Recommendations to Prevent Overfeeding. Nutrients. 2023 Sep 15;15(18):4004. doi: 10.3390/nu15184004.

Salgin B, Norris SA, Prentice P, Pettifor JM, Richter LM, Ong KK, Dunger DB. Even transient rapid infancy weight gain is associated with higher BMI in young adults and earlier menarche. Int J Obes (Lond). 2015 Jun;39(6):939-44. doi: 10.1038/ijo.2015.25.

Flores-Barrantes P, Iguacel I, Iglesia-Altaba I, Moreno LA, Rodríguez G. Rapid Weight Gain, Infant Feeding Practices, and Subsequent Body Mass Index Trajectories: The CALINA Study. Nutrients. 2020 Oct 17;12(10):3178. doi: 10.3390/nu12103178.

Wood CT, Witt WP, Skinner AC, Yin HS, Rothman RL, Sanders LM, Delamater AM, Flower KB, Kay MC, Perrin EM. Effects of Breastfeeding, Formula Feeding, and Complementary Feeding on Rapid Weight Gain in the First Year of Life. Acad Pediatr. 2021 Mar;21(2):288-296. doi: 10.1016/j.acap.2020.09.009.

Specific comments

Introduction, Line 31-33

Literature citations showing protective effects of breastfeeding against obesity misses important papers that might be included:

Rajamoorthi A, LeDuc CA, Thaker VV. The metabolic conditioning of obesity: A review of the pathogenesis of obesity and the epigenetic pathways that "program" obesity from conception. Front Endocrinol (Lausanne). 2022 Oct 18;13:1032491. doi: 10.3389/fendo.2022.1032491.

Woo JG, Martin LJ. Does Breastfeeding Protect Against Childhood Obesity? Moving Beyond Observational Evidence. Curr Obes Rep. 2015 Jun;4(2):207-16. doi: 10.1007/s13679-015-0148-9.

Horta BL, Loret de Mola C, Victora CG. Long-term consequences of breastfeeding on cholesterol, obesity, systolic blood pressure and type 2 diabetes: a systematic review and meta-analysis. Acta Paediatr. 2015 Dec;104(467):30-7. doi: 10.1111/apa.13133. PMID: 26192560.

There is also a deficiency of important references supporting the diabetes-preventive effect of breastfeeding.

Owen CG, Martin RM, Whincup PH, Smith GD, Cook DG. Does breastfeeding influence risk of type 2 diabetes in later life? A quantitative analysis of published evidence. Am J Clin Nutr. 2006 Nov;84(5):1043-54. doi: 10.1093/ajcn/84.5.1043. Erratum in: Am J Clin Nutr. 2012 Mar;95(3):779.

Pettitt DJ, Forman MR, Hanson RL, Knowler WC, Bennett PH. Breastfeeding and incidence of non-insulin-dependent diabetes mellitus in Pima Indians. Lancet. 1997 Jul 19;350(9072):166-8. doi: 10.1016/S0140-6736(96)12103-6.

Young TK, Martens PJ, Taback SP, Sellers EA, Dean HJ, Cheang M, Flett B. Type 2 diabetes mellitus in children: prenatal and early infancy risk factors among native canadians. Arch Pediatr Adolesc Med. 2002 Jul;156(7):651-5. doi: 10.1001/archpedi.156.7.651.

Horta BL, de Lima NP. Breastfeeding and Type 2 Diabetes: Systematic Review and Meta-Analysis. Curr Diab Rep. 2019 Jan 14;19(1):1. doi: 10.1007/s11892-019-1121-x.

Melnik BC, Weiskirchen R, Weiskirchen S, Stremmel W, John SM, Leitzmann C, Schmitz G. Diabetes-preventive molecular mechanisms of breast versus formula feeding: new insights into the impact of milk on stem cell Wnt signaling. Front Nutr. 2025 Jul 29;12:1652297. doi: 10.3389/fnut.2025.1652297.

Line 34

The authors state that formula feeding provides “adequate nutrition”.

Apparently, this is not the case as early growth trajectories and obesity risk in adolescence deviate from physiological breastfeeding patterns.

In contrast to artificial formula, breastmilk executes adequate postnatal programming under surveillance of the human lactation genome and provides both sufficient nutrient supply and oral programming by bioactive signaling compounds of mother´s mammary glands.

Author Response

Reviewer’s comment 1:
General comments

The investigators analyzed secondary data from the 2023 National Growth Survey on Preschool Children in Japan. Anthropometric measurements were obtained at a 1-month postnatal checkup or at checkups arranged for the survey. Breastfed infants were smaller in length/height and weight than formula-fed peers during the first 2 years, with the largest differences in infancy. The investigators showed that breastfed Japanese children are smaller in early life but achieve catch-up growth by 5 years.

In the paper´s introduction the authors discuss their concern of increased risk of obesity by formula feeding associated with rapid postnatal growth as opposed to breastfeeding. In the discussion, they cite a recent Japanese study showing that formula-fed children have a higher prevalence of obesity at 15 years of age than those who were breastfed.

Kadowaki T, Matsumoto N, Suzuki E, Mitsuhashi T, Takao S, Yorifuji T. Breastfeeding at 6 months of age had a positive impact on overweight and obesity in Japanese adolescents at 15 years of age. Acta Paediatr. 2023 Jan;112(1):106-114. doi: 10.1111/apa.16551.

However, the discussion does not provide any idea how their findings might relate formula feeding to an increased risk of obesity. Obviously, crude anthropometric data like weight measurement does not provide answers for body mass composition, distribution of lean and fat mass, relative distribution of white and beige/brown adipose tissue and white/brown adipose tissue stem cell fate decisions.
The authors could improve their DISCUSSION considering that formula feeding is associated with higher blood levels of IGF-1, which accelerates postnatal growth (rapid weight gain).

Socha P, Hellmuth C, Gruszfeld D, Demmelmair H, Rzehak P, Grote V, Weber M, Escribano J, Closa-Monasterolo R, Dain E, Langhendries JP, Riva E, Verduci E, Koletzko B; European Childhood Obesity Trial Study Group. Endocrine and Metabolic Biomarkers Predicting Early Childhood Obesity Risk. Nestle Nutr Inst Workshop Ser. 2016;85:81-8. doi: 10.1159/000439489.

Importantly, IGF-1 also promotes adipogenesis by a lineage bias of endogenous adipose stem/progenitor cells. In fact, IGF-1 attenuates Wnt/β-catenin signaling by activating Axin2/PPARγ pathways in cells of the stromal vascular fraction.

Hu L, Yang G, Hägg D, Sun G, Ahn JM, Jiang N, Ricupero CL, Wu J, Rodhe CH, Ascherman JA, Chen L, Mao JJ. IGF1 Promotes Adipogenesis by a Lineage Bias of Endogenous Adipose Stem/Progenitor Cells. Stem Cells. 2015 Aug;33(8):2483-95. doi: 10.1002/stem.2052.

Thus, IGF-1-mediated acceleration of growth by formula feeding may be an indirect indicator of IGF-1-driven extension of adipocyte stem cells and lifelong obesity risk.

A further limitation of this study that should be mentioned is the missing information of total daily protein intake by the three feeding groups, as the amount of protein intake determines circulatory IGF-1 levels and FTO expression controlling both growth trajectories and adipogenesis.

Melnik BC, Weiskirchen R, John SM, Stremmel W, Leitzmann C, Weiskirchen S, Schmitz G. White Adipocyte Stem Cell Expansion Through Infant Formula Feeding: New Insights into Epigenetic Programming Explaining the Early Protein Hypothesis of Obesity. Int J Mol Sci. 2025 May 8;26(10):4493. doi: 10.3390/ijms26104493.

The reviewer wonders why the authors did not discuss recent publications on rapid weight gain by formula feeding and

Appleton J, Russell CG, Laws R, Fowler C, Campbell K, Denney-Wilson E. Infant formula feeding practices associated with rapid weight gain: A systematic review. Matern Child Nutr. 2018 Jul;14(3):e12602. doi: 10.1111/mcn.12602.

Dharod JM, McElhenny KS, DeJesus JM. Formula Feeding Is Associated with Rapid Weight Gain between 6 and 12 Months of Age: Highlighting the Importance of Developing Specific Recommendations to Prevent Overfeeding. Nutrients. 2023 Sep 15;15(18):4004. doi: 10.3390/nu15184004.

Salgin B, Norris SA, Prentice P, Pettifor JM, Richter LM, Ong KK, Dunger DB. Even transient rapid infancy weight gain is associated with higher BMI in young adults and earlier menarche. Int J Obes (Lond). 2015 Jun;39(6):939-44. doi: 10.1038/ijo.2015.25.

Flores-Barrantes P, Iguacel I, Iglesia-Altaba I, Moreno LA, Rodríguez G. Rapid Weight Gain, Infant Feeding Practices, and Subsequent Body Mass Index Trajectories: The CALINA Study. Nutrients. 2020 Oct 17;12(10):3178. doi: 10.3390/nu12103178.

Wood CT, Witt WP, Skinner AC, Yin HS, Rothman RL, Sanders LM, Delamater AM, Flower KB, Kay MC, Perrin EM. Effects of Breastfeeding, Formula Feeding, and Complementary Feeding on Rapid Weight Gain in the First Year of Life. Acad Pediatr. 2021 Mar;21(2):288-296. doi: 10.1016/j.acap.2020.09.009.

Authors' response 1:

We appreciate the reviewer’s thoughtful comments regarding mechanistic pathways and the subsequent risk of obesity. Our analysis used a single-wave, cross-sectional dataset with one anthropometric observation in each child (plus birth information). Therefore, the survey did not include endocrine markers (e.g., IGF-1), detailed dietary composition (e.g., protein intake), or body composition measures. Because of these constraints, the present study cannot test IGF-1-related mechanisms, evaluate the “early-protein” hypothesis, or quantify the causal risk for later adiposity.

To avoid over-interpretation, we have (i) clarified in the Introduction section that our scope is age-specific cross-sectional anthropometry, (ii) added a brief, neutral acknowledgement to the Discussion section stating that hypothesized associations between feeding patterns, rapid weight gain, and endocrine/nutritional pathways cannot be evaluated by our data, and (iii) stated explicitly in the Strengths and limitations section that mechanistic and long-term metabolic inferences are beyond the scope of this analysis.

Changes to the manuscript
Introduction

“Commercial infant formula is designed to meet infant nutrient requirements when breastfeeding is not possible. Nevertheless, previous studies have reported that formula is associated with more rapid somatic growth in early infancy and a higher risk of overweight or obesity than breastfeeding [7-11].”

Discussion (Main findings)

“Previous studies have suggested a potential association between formula feeding and an increased risk of later obesity [2,9-12]. In Japan, cohort data have indicated that formula-fed children may show a higher prevalence of obesity at 15 years of age than those who are breastfed [29]. Infant formula generally contains more protein than human milk, and this higher protein intake has been suggested to increase the risk of rapid weight gain during early infancy [11,30]. Rapid weight gain in infancy associated with an increased risk of overweight and obesity [10,31]. Moreover, formula feeding might be associated with increased body mass index in school-age children, independent of whether rapid weight gain occurs [10]. In our study, we did not assess adiposity or metabolic outcomes, and no causal inference can be made.”

Discussion (Strengths and limitations)

“Endocrine markers, dietary macronutrient information (including protein), and body composition measurements were not collected. Therefore, mechanistic explanations and inferences regarding the risk of later obesity are beyond the scope of this study.”

Reviewer’s comment 2:
Specific comments
Introduction, lines 31–33
Literature citations showing protective effects of breastfeeding against obesity misses important papers that might be included:

Rajamoorthi A, LeDuc CA, Thaker VV. The metabolic conditioning of obesity: A review of the pathogenesis of obesity and the epigenetic pathways that "program" obesity from conception. Front Endocrinol (Lausanne). 2022 Oct 18;13:1032491. doi: 10.3389/fendo.2022.1032491.

Woo JG, Martin LJ. Does Breastfeeding Protect Against Childhood Obesity? Moving Beyond Observational Evidence. Curr Obes Rep. 2015 Jun;4(2):207-16. doi: 10.1007/s13679-015-0148-9.

Horta BL, Loret de Mola C, Victora CG. Long-term consequences of breastfeeding on cholesterol, obesity, systolic blood pressure and type 2 diabetes: a systematic review and meta-analysis. Acta Paediatr. 2015 Dec;104(467):30-7. doi: 10.1111/apa.13133. PMID: 26192560.

There is also a deficiency of important references supporting the diabetes-preventive effect of breastfeeding.

Owen CG, Martin RM, Whincup PH, Smith GD, Cook DG. Does breastfeeding influence risk of type 2 diabetes in later life? A quantitative analysis of published evidence. Am J Clin Nutr. 2006 Nov;84(5):1043-54. doi: 10.1093/ajcn/84.5.1043. Erratum in: Am J Clin Nutr. 2012 Mar;95(3):779.

Pettitt DJ, Forman MR, Hanson RL, Knowler WC, Bennett PH. Breastfeeding and incidence of non-insulin-dependent diabetes mellitus in Pima Indians. Lancet. 1997 Jul 19;350(9072):166-8. doi: 10.1016/S0140-6736(96)12103-6.

Young TK, Martens PJ, Taback SP, Sellers EA, Dean HJ, Cheang M, Flett B. Type 2 diabetes mellitus in children: prenatal and early infancy risk factors among native canadians. Arch Pediatr Adolesc Med. 2002 Jul;156(7):651-5. doi: 10.1001/archpedi.156.7.651.

Horta BL, de Lima NP. Breastfeeding and Type 2 Diabetes: Systematic Review and Meta-Analysis. Curr Diab Rep. 2019 Jan 14;19(1):1. doi: 10.1007/s11892-019-1121-x.

Melnik BC, Weiskirchen R, Weiskirchen S, Stremmel W, John SM, Leitzmann C, Schmitz G. Diabetes-preventive molecular mechanisms of breast versus formula feeding: new insights into the impact of milk on stem cell Wnt signaling. Front Nutr. 2025 Jul 29;12:1652297. doi: 10.3389/fnut.2025.1652297.

Authors’ comment 2:

We are grateful to the reviewer for the suggested references. However, because we analyzed cross-sectional anthropometry up to 60 months and did not include metabolic outcomes, we wish to avoid over-interpretation of our data. Accordingly, we have kept the text unchanged and addressed the reviewer’s comment by adding one comprehensive, recent systematic review to the Introduction and Discussion sections.

Patnode CD, Henrikson NB, Webber EM, Blasi PR, Senger CA, Guirguis-Blake JM. “Breastfeeding and Health Outcomes for Infants and Children: A Systematic Review.” Pediatrics. 2025;156. doi:10.1542/peds.2025-071516.

Reviewer’s comment 3:
Line 34

The authors state that formula feeding provides “adequate nutrition”. Apparently, this is not the case as early growth trajectories and obesity risk in adolescence deviate from physiological breastfeeding patterns.

In contrast to artificial formula, breastmilk executes adequate postnatal programming under surveillance of the human lactation genome and provides both sufficient nutrient supply and oral programming by bioactive signaling compounds of mother´s mammary glands.

Authors’ comment 3:

We thank the reviewer for the comment. To avoid implying a judgment about nutritional adequacy and to remain aligned with our study’s scope, we have revised this sentence to provide a neutral, evidence-based view and have cited a comprehensive systematic review.

Changes to the manuscript
Introduction

“Commercial infant formula is designed to meet infant nutrient requirements when breastfeeding is not possible. Nevertheless, previous studies have reported that formula is associated with more rapid somatic growth in early infancy and a higher risk of overweight or obesity than breastfeeding [7–11].”

Round 2

Reviewer 2 Report

Comments and Suggestions for Authors

I commend the authors on a thorough and compelling revision. Well done! I have no further concerns about the manuscript. It was a pleasure to review this revision.